# Polyhydroxyalkanoate Decelerates the Release of Paclitaxel from Poly(lactic-co-glycolic acid) Nanoparticles

**DOI:** 10.3390/pharmaceutics14081618

**Published:** 2022-08-02

**Authors:** Si Yeong Lee, So Yun Kim, Sook Hee Ku, Eun Ji Park, Dong-Jin Jang, Sung Tae Kim, Seong-Bo Kim

**Affiliations:** 1Department of Nanoscience and Engineering, Inje University, Gimhae 50834, Korea; leesyeong98@naver.com (S.Y.L.); soyun1598@naver.com (S.Y.K.); 2CJ CheilJedang, Suwon 16495, Korea; sookhee.ku@cj.net; 3College of Pharmacy, Inje University, Gimhae 50834, Korea; ghkswls05@naver.com; 4Department of Bio-Health Technology, College of Biomedical Science, Kangwon National University, Chunchoen 24341, Korea; djjang@kangwon.ac.kr; 5Department of Pharmaceutical Engineering, Inje University, Gimhae 50834, Korea; 6Bio-Living Engineering Major, Global Leaders College, Yonsei University, Seoul 03722, Korea

**Keywords:** Poly(lactide co-glycolide), polyhydroxyalkanoate, nanoparticle, drug release, paclitaxel

## Abstract

Biodegradable nanoparticles (NPs) are preferred as drug carriers because of their effectiveness in encapsulating drugs, ability to control drug release, and low cytotoxicity. Although poly(lactide co-glycolide) (PLGA)-based NPs have been used for controlled release strategies, they have some disadvantages. This study describes an approach using biodegradable polyhydroxyalkanoate (PHA) to overcome these challenges. By varying the amount of PHA, NPs were successfully fabricated by a solvent evaporation method. The size range of the NPS ranged from 137.60 to 186.93 nm, and showed zero-order release kinetics of paclitaxel (PTX) for 7 h, and more sustained release profiles compared with NPs composed of PLGA alone. Increasing the amount of PHA improved the PTX loading efficiency of NPs. Overall, these findings suggest that PHA can be used for designing polymeric nanocarriers, which offer a potential strategy for the development of improved drug delivery systems for sustained and controlled release.

## 1. Introduction

Biodegradable polymers have a variety of applications in the pharmaceutical field [1]. These polymers, either natural or synthetic, provide desirable physicochemical properties for controlled release of therapeutic agents, with concomitant improvements to their pharmacokinetics [2,3]. Biodegradable polymers have been applied to drug delivery systems in many previous studies [4,5]. However, there remains resistance to their use, and concerns regarding their degradation [6]. Degraded metabolites can accumulate and cause local toxic responses. For example, during degradation, metabolites triggered undesirable toxicity and pro-inflammatory mediators in the microenvironment, despite having high biocompatibility [7].

Pharmaceutical formulations have been developed to control the release profiles of therapeutics, which is important for alleviating toxicity and achieving improved pharmacokinetics, as well as therapeutic efficacy [8]. Various polymers, such as poly-ε-caprolactone (PCL), poly(l-lactide) (PLA), and poly(lactide co-glycolide) (PLGA), have been widely explored in previous studies as important nanoparticle wall materials for controlling the release of therapeutics [9], the rate of which is mainly dependent on the diffusion of therapeutics through a matrix/barrier composed of different polymers [10]. However, PLGA-based matrices, the most prevalent biodegradable copolymer for drug delivery carriers, have a few drawbacks. These include a relatively low encapsulation efficiency and an initial burst of drug release [11,12]. This initial burst of drug is undesirable because it can cause toxicity and safety issues, owing to excessive amounts of the drug in the early stages after administration [13]. Therefore, there is a need to reduce the initial burst of drug release and to further control the release through modified polymeric formulations. 

Polyhydroxyalkanoates (PHAs) are biopolymers synthesized inside microorganisms for carbon and energy storage [14]. PHAs, which are natural polyesters, have many advantages, such as biocompatibility, biodegradability, and high loading efficacy, and are suitable for a variety of biomedical and pharmaceutical applications [15]. Their degraded monomers, such as 3-hydroxybutyrate and 4-hydroxybutyrate, are easily metabolized and excreted well [16]. These advantages make them attractive substitutes for biodegradable polymers as drug delivery carriers [17]. Previous studies have reported PHA-based strategies for encapsulating therapeutics [18] and cells [19], which implies the potential of PHAs as pharmaceutical materials and as alternatives to other biopolymers for industrial use. In addition, their tunable physicochemical properties are useful and can be controlled through the production process [20]. However, the main hurdles against the use of PHA are their high hydrophobicity [21] and high cost for conventional production [22,23]. If these limitations are overcome, PHAs will be more attractive and useful as pharmaceutical excipients. As with conventional biopolymers, controlling the release profiles of therapeutics is also advantageous for PHA-based carriers, although their degradability, stiffness, and elasticity are tunable [24]. Such controlled release of therapeutics via PHA matrices will provide a wide range of applications for drug delivery and biomedical engineering. As a therapeutic agent, paclitaxel (PTX), C_47_H_51_NO_14_ (CAS NO. 33069-62-4, MW 853.906 g/mol), was chosen because it has been widely used as a chemotherapeutic agent for treatment of patients with lung, ovarian, breast, head, and neck cancer [25]. Due to its hydrophobicity and toxicity, PTX requires a pharmaceutical formulation, leading to an improved colloidal stability and controlled release properties. In practice, biocompatible polymers, such as PLGA, PLA, and/or PGA, have been applied for clinical use [26]. Despite these efforts, there remain some limitations for the sustained release of PTX. 

The goal of this study was to fabricate polymeric nanoparticles (NPs) composed of PLGA and PHA to control the release of paclitaxel (PTX), a model therapeutic agent, and to improve drug release and entrapment by varying the composition of the wall materials. 

## 2. Materials and Methods

### 2.1. Materials

PLGA (50/50, dl-lactide/glycolide copolymer, 70,000–150,000 g/mol) (Boehringer Ingelheim-Korea, Seoul, Republic of Korea) and PTX (Samyang Biopharmaceuticals, Seongnam, Republic of Korea) were supplied by Prof. Dong-Jin Jang at Kangwon National University. PHA was supplied by CJ CheilJedang (Seoul, Republic of Korea). Polyvinyl alcohol (PVA), a surfactant, was purchased from Daejung Chemicals and Metals (Daejung, Siheung, Republic of Korea); chloroform, an oil phase of emulsion, was obtained from Duksan General Science (Duksan, Ansan, Republic of Korea) for the solvent evaporation process. All other reagents used were of analytical grade, and double-distilled water was used.

### 2.2. Preparation of Polymeric NPs

NPs composed of PLGA and/or PHA were fabricated by the two-step solvent evaporation method [27,28]. In brief, 10 mg of the polymer was first dissolved in 2 mL of chloroform as an oil phase, which was added to an aqueous phase containing 3% PVA. To obtain NPs, the mixture was first homogenized under the following conditions: 300 watts for 1 min in an ice bath (Scientz-11D, NingBo Scientz Biotechnology, Ningbo, Zhejiang, China). In the second step, these emulsions were poured into a 0.3% PVA solution. To completely evaporate the solvent, each emulsified mixture was stirred at 500 rpm (MS-MP-8, Daihan Scientific, Daegu, Republic of Korea) for 3 h in a fume hood. Then, polymeric NPs (PLGA, PHA, and PLGA-PHA mixed matrices) were harvested by centrifugation at 24,249 g for 3 h (ScanSpeed 1580R, Gyrozen, Gimpo, Republic of Korea). These NPs were washed by double-distilled water to eliminate the remnant PVA, and retained for further studies. In addition, the entrapment efficiency of PTX was determined by calculating the ratio of the amount of incorporated PTX in the NPs to the total amount of PTX in the NPs. Sedimented NPs were dissolved by dimethyl sulfoxide and vortexed, thereafter quantitatively analyzed by high-performance liquid chromatography (HPLC), which is further explained in Section 2.5. 

### 2.3. Physicochemical Properties of Polymeric NPs

#### 2.3.1. Size and Zeta Potential Measurement of NPs 

The size of NPs was measured via dynamic light scattering using Zetasizer (Nano-ZS90, Malvern Instruments, Worcestershire, UK) at a fixed angle of 90° at 25 °C. All samples were washed with double-distilled water to remove PVA. Measurements were performed in triplicate after dispersion of NPs in double-distilled water. Each zeta potential value of NPs was investigated by zeta potential measurement at 25 °C. Data are presented as the mean ± standard deviation.

#### 2.3.2. Fourier-Transform Infrared Spectroscopic Measurement of NPs 

Polymeric NPs composed of PLGA, PHA, and PLGA-PHA mixed matrices were fabricated using the solvent evaporation method and harvested as described above. Each sample was placed on an attenuated total reflection (ATR) crystal, and spectra were scanned between 4000 and 500 cm^−1^, including 64 scans with a resolution of 4 cm^−1^ using a Fourier transform infrared (FT-IR) spectrometer (ALPHA II, Bruker, Karlsruhe, Germany) [29]. Spectral peaks were confirmed by transmittance [30], and each peak of these polymers was comparatively analyzed for a better understanding of shell materials. 

#### 2.3.3. Differential Scanning Calorimetric Measurement of NPs 

Differential scanning calorimetry (DSC) was used to analyze various physical properties and thermal transitions of polymeric materials. In brief, each harvested sample was measured using DSC equipment (DSC-60, Shimadzu, Kyoto, Japan) [31]. Approximately 5–7 mg of each sample was placed in a sealed aluminum pan prior to heating under air flow (20 mL/min) and then heated at a rate of 5 °C/min. DSC scans of NPs were obtained from 20 °C to 230 °C.

#### 2.3.4. X-ray Diffraction Measurement of NPs 

The X-ray diffraction patterns of NPs were obtained using an X-ray diffraction system equipped with Cu-Kα irradiation at 40 kV (Ultima IV, Rigaku, Tokyo, Japan). The diffractogram was obtained in the range of 2–80°/2θ at a step of 1°/min [32,33].

### 2.4. Storage Stability of NPs

NPs fabricated as mentioned in Section 2.2 were stored at 4 °C in a cold chamber, and their size and zeta potential (ζ) values were measured by dynamic light scattering using a Zetasizer every day for one week. Prior to measurement, each NP was washed twice with double-distilled water to eliminate the remnant PVA.

### 2.5. In Vitro Release of PTX from NPs 

Prior to the release experiment, PTX, used as a model drug, was quantitatively analyzed by HPLC using Hitachi LaChrom HPLC-2000 series equipped with an autosampler (L-2200), pump (L-2130), column oven (L-2300), and diode array detector (L-2455) (Hitachi Co., Ltd., Tokyo, Japan). An ODS-2 C_18_ column (150 × 4.6 mm, 5 μm; LB Science, Seoul, Republic of Korea) was used for quantitative analysis. The mobile phase was composed of acetonitrile and water (50:50, *v*/*v*) at a flow rate of 0.8 mL/min and injection volume was set as 10 μL. Each sample was measured at 227 nm and analyzed using the D-2000 Elite software [34]. 

In vitro release experiments were performed using the modified dialysis membrane method [35,36]. In brief, harvested PTX-loaded NPs were added into a Slide-A-Lyzer^®^ dialysis cassette, with a 10,000 molecular weight cut-off membrane (Thermo Fisher Scientific, Rockford, IL, USA), which is sufficient to allow PTX to penetrate the membrane during dialysis. Each cassette containing PTX-loaded NPs was placed into bottles filled with 30 mL of phosphate buffer at pH 7.4, which were then placed in a shaking incubator (100 rpm) (JSSI-100T, JSR Research Inc., Gonju, Republic of Korea) at 37 °C. Samples were then harvested from the cassette at various release times, 7 h after administration. One milliliter of sample harvested at each time point was quantitatively analyzed as described above and replaced with the same amount of buffer. 

### 2.6. In Vitro Cell Experiment 

HeLa cells obtained from the Korean Cell Line Bank (KCLB No. 1002) were cultured in Dulbecco’s Modified Eagle’s Medium (Corning, VA, USA) supplemented with 10% fetal bovine serum (Corning, VA, USA) and 1% penicillin-streptomycin (Mediatech Inc., Corning, VA, USA). Cells were maintained at 37 °C under humified conditions with 5% CO_2_ (ARA150 CO_2_ incubator, Gyrozen Co., Ltd., Gimpo, Republic of Korea). For confocal microscopy, HeLa cells (1.6 × 10^5^) were seeded in microscopy chambers with coverslip bottoms (Marienfeld, Harsewinkel, Germany). After stabilization, the cells were incubated for 4 h with polymeric NPs containing PTX and Nile red (<0.1 mg/mL, Tokyo Chemical Industry, Tokyo, Japan) [37,38]. During incubation, NPs were treated with 100 nM of lysotracker (LysoTracker^TM^ Green DND-26, Invitrogen, OR, USA) for 2 h and mounting medium was treated with 4,6-diamidino-2-phenylindole (DAPI) (Fluoroshield^TM^ with DAPI, Sigma, St. Louis, MO, USA). NPs were observed by confocal microscopy (Carl Zeiss-LSM800, Zeiss, Jena, Germany) (DAPI: 465 nm, LysoTracker: 509 nm, Nile red: 636 nm, 63×, ZEN2.6).

## 3. Results

### 3.1. Physicochemical Properties of NPs

#### 3.1.1. Size and Zeta Potential of NPs

NPs composed of PLGA and/or PHA were fabricated, and their physicochemical characteristics were analyzed, as shown in Table 1. The size of the NPs was less than 200 nm, which were well-matched with the transmission electron microscopy (TEM) images (Appendix A). According to previous literature, NPs have appropriate sizes for tumor accumulation, as particles larger than 200 nm are mostly eliminated by the phagocytic system during circulation [39,40]. The polydispersity index (PDI) value of the NPs was very low (<0.25), indicating uniformity of the NPs [41]. The zeta potential (ζ) value of NP1 to NP4 was slightly negative, presumably providing a stable dispersion, owing to the electrostatic repulsion between NPs in a given experiment. Based on the physicochemical properties, it was considered that the NPs were well fabricated for passive targeting of antineoplastic agents, although their size varied depending on the polymer composition. Notably, NP4, composed of PHA, showed a higher entrapment efficiency of PTX than that of NP1, composed of PLGA. As the amount of PHA increased, entrapment efficiency also gradually increased, which might have improved the therapeutic effects and reduced side effects [42].

#### 3.1.2. Fourier-Transform Infrared (FT-IR) Spectroscopic Analysis 

The chemical structure of the NPs was analyzed by FT-IR spectroscopy, as shown in Figure 1. Identification of IR spectra provides information about different functional groups of PLGA and/or PHA polymers in the NPs, compared to those of raw materials, such as PLGA and PHA (Appendix A). The NP1 spectrum showed intense spectra that belonged to PLGA from NP1, which exhibited molecular vibration of the representative functional groups, including aliphatic C-H stretching vibration (3010 to 2885 cm^−1^), C=O stretching vibration (1760 to 1750 cm^−1^), and C-O ester stretching vibration (1300 to 1150 cm^−1^) [43,44]. The strong peak at 1756.34 cm^−1^ corresponded to a carbonyl group stretch in lactide and glycolide. The peak at 1422.45 cm^−1^ corresponded to CH_2_ bending of glycolide, and the other peak at 1088.01 cm^−1^ corresponded to C-O stretching vibration [45]. In contrast, the NP4 spectrum showed typical PHA characteristics. The representative band at 1732.34 cm^−1^ corresponded to a carbonyl (C=O) ester bond stretching vibration related to the amorphous region of PHA [46]. The absorption band at 2916.44 cm^−1^ corresponded to an asymmetric stretching vibration of CH_2_ associated with the formation of lateral chains of monomers [47]. Other peaks in the range of 1450 to 1000 cm^−1^ represented the functional groups of CH_3_, CH_2_, and C-O, and in the range of 1300 to 1000 cm^−1^ represented stretching of the ethers of C-O-C linkage [48]. The IR spectra of NPs (NP2 and NP3) composed of PLGA and PHA mixed matrices were analyzed and compared with those of NP1 and NP4. The spectra of NP2 and NP3 showed common characteristics of both PLGA and PHA. For example, strong bands in NP2 and NP3 spectra in the range of 1740 to 1724 cm^−1^ corresponding to C=O stretching vibration was a typical feature of PHA [49,50]. Both NPs also showed typical peaks that were similar to those of PLGA at approximately 1422 cm^−1^ and 1088 cm^−1^. These findings suggest no significant interaction between PLGA and PHA.

#### 3.1.3. Differential Scanning Calorimetric (DSC) Analysis

DSC thermal analysis of each NP composed of PLGA and/or PHA was conducted as described in Section 2.3.3. The DSC thermograms (Figure 2 and Appendix A) showed glass transition temperatures. As shown in Figure 2, the endothermic peak in NP1 was observed at 46.66 °C, corresponding to the glass transition temperature (Tg), which was consistent with the Tg of PLGA [51]. The Tg values of NP2 and NP3 were slightly shifted in accordance with the ratio of PHA to PLGA, presumably because of the increased amount of PHA [52]. There were no peaks related to crystallinity or melting temperature, indicating an amorphous state.

#### 3.1.4. X-ray Diffraction (XRD) Analysis of NPs

The physical state of NPs was studied using XRD crystallography, as shown in Figure 3 and Appendix A. The XRD patterns of the NPs composed of PLGA and PHA were analyzed and compared. All NPs showed no significant peaks indicative of crystallinity in a given range of 2θ, which was consistent with the DSC analysis, and there were no remarkable peaks corresponding to the crystallization temperature in any given experiment. The XRD patterns of NPs showed a broad band in the following regions: NP1 (9.49° to 35.56°), NP2 (9.28° to 36.91°), NP3 (9.01° to 36.89°), and NP4 (8.74° to 38.07°). As shown in the XRD patterns of NP2 and NP3, the mixed composition of PHA and PLGA rarely affected the amorphous state of the NPs, although there was a slight change in the region of the broad band.

### 3.2. Storage Stability of NPs

The NPs were stored at 4 °C, and their size and zeta potential values were measured to evaluate particle stability, as shown in Figure 4. None of the NPs showed any significant change with respect to the size during the given period. The following changes in the size of NPs were less than 10%: NP1 (<8.41%), NP2 (<6.18%), NP3 (<4.99%), and NP4 (<9.23%), respectively. Although there were slight changes in the zeta potential values, all NPs remained negatively charged (Figure 4b and Appendix A), which provides efficient electrostatic repulsion [53].

### 3.3. In Vitro Release of PTX from NPs

In vitro release studies of PTX from NPs were performed under physiological conditions (pH 7.4) [54]. As shown in Figure 5, PTX was gradually released over time from the polymeric matrices of NPs during the given period; however, the release pattern was significantly different depending on the polymer composition. During the early period (0–90 min), NP1 composed of PLGA showed a slightly higher PTX release than that of other NPs; however, the total release was less than 10% in pH 7.4, indicating that the addition of PHA might have reduced PTX release in NP2, NP3, and NP4. After 7 h, NP1 showed 92.43% release, whereas NP4, composed of PHA, showed 43.90% release of PTX. Under the same conditions, NP2 released 68.65% of PTX, indicating a reduced release of PTX. The PTX released from NP 4 was not statistically different from the PTX release from NP3. The tendency of reduced release of PTX was observed throughout the 7 h duration of the experiment. Overall, the findings demonstrated that the addition of PHA decelerated the release of PTX at pH 7.4. 

Based on PTX release data, the release curves from Figure 5 were plotted to represent the following mathematical models: zero-order, first-order, Higuchi, and Hixon–Crowell kinetic models. As shown in Table 2 and Appendix A, all NPs exhibited a high correlation with the zero-order kinetic model (correlation coefficient, R^2^ > 0.97) rather than the Higuchi kinetic model at pH 7.4. The decreased K value was correlated with the reduced release of PTX, as shown in Table 2. The results are consistent with PTX being released in a zero-order manner for up to 7 h at physiological pH. The release of PTX at close to zero-order kinetics could overcome problems of immediate release/initial burst release, as a constant rate of drug release was observed without any significant changes, resulting from the addition of PHA.

### 3.4. Intracellular Uptake of NPs

LysoTracker staining of acidic compartments, such as lysosomes, was performed to determine the intracellular uptake mechanism of NPs. Prior to the experiment, NPs were stained with the fluorescent dye Nile red and added to HeLa cells for incubation. As shown in Figure 6, NPs were delivered to the cytoplasm (red color) rather than the nuclei (blue color) of HeLa cells. Most NPs (NP1–NP4) colocalized with lysosomes (green color) without any significant differences among NPs, as shown in Figure 6 (merged). Although there were slight differences in size and zeta potential values among NPs (Table 1), most NPs appeared to be endocytosed in HeLa cells, based on the evidence of colocalization of NPs with lysosomes (merged) [55]. Taking these results together, NPs were successfully delivered to the cells independent of the polymer composition.

## 4. Discussion

Biodegradable biocompatible polymer-based NPs have been widely studied for biomedical and pharmaceutical applications [56,57]. NPs can improve stability and enhance the solubility of entrapped cargos carrying therapeutic agents [58]. Notably, the use of biocompatible NPs reduces the concomitant toxic effect of polymeric materials and is beneficial for controlled delivery of therapeutics [59]. The use of such NPs as drug delivery carriers has also attracted great interest because of their biocompatibility and biosafety [60]. PLGA is a biodegradable, biocompatible polymer that is capable of encapsulation of various therapeutic agents and is suitable for controlled release systems [61]. However, despite its safety and approval by the United States Food and Drug Administration, the use of PLGA has been limited, owing to its low entrapment efficiency and difficulty in regulating drug release [62]. In this study, biodegradable PHA, which is produced by microbes, was used to improve entrapment efficacy and regulate the release profile of drugs, depending on the polymer composition. Although a few approaches using PHA have been described, simultaneous use of PLGA and PHA as core materials has not previously been reported [18,19].

The results presented here demonstrated that the physical properties (particle size and zeta potential) of NPs were slightly influenced by the addition of PHA; however, all fabricated NPs were smaller than 200 nm and retained negative surface charges (Table 1). Previous literature reported that small-sized NPs (less than 10 nm) were rapidly excreted by the kidneys, whereas large-sized NPs (more than 200 nm) showed a risk of activating the complement system in the body [58,63]. Thus, the fabricated NPs were considered to have an appropriate size range to be used as antitumor drug carriers. In addition, a previous study reported that neutral or slightly negatively charged NPs showed a longer circulation time, whereas positively charged NPs were rapidly cleared [64]. Such anionic NPs generally show fewer interactions with biomolecules, leading to less impact of the protein corona phenomena because most proteins are negatively charged in the physiological environment [65]. In this aspect, it is considered that anionic surfaces, such as those of the NPs described here, bind to a small amount of cationic proteins before reaching the target region. The protein corona effect was clearly observed in a previous study from this group [66]. In general, anionic NPs show relatively lower toxicity than cationic NPs, which disturb the cellular membrane [67]. Moreover, unlike anionic NPs, cationic NPs can trigger immunological responses [68] and concomitant cytotoxicity depending on the surface functionalities, which was also observed in a previous study [69]. In summary, previous reports have demonstrated that anionic surface properties and an appropriate size of NPs (<200 nm), such as those of the NPs fabricated in this study, have several benefits, such as a longer circulation time, lower protein corona, reduced cytotoxicity, and reduced immune response. 

Furthermore, the use of PHA increased the entrapment efficiency of PTX compared to NP1 composed of PLGA alone (Table 1) and regulated the release of PTX, without any significant changes in the release kinetics (Figure 5, Table 2). Moreover, the NPs described here showed zero-order release of PTX, and the addition of PHA reduced the release for at least up to 7 h. This zero-order release is a potential kinetic pattern of drugs to overcome the drawbacks, such as immediate release and first-order kinetics of PLGA-based NPs, and is quite suitable for sustained release [70]. Based on these results, the findings suggest that adjusting the polymer composition could improve these advantages and compensate for the disadvantages of PLGA-based NPs resulting from the use of PHA (Figure 1). Furthermore, as shown in Figure 6, intracellular uptake of NPs was almost independent of the polymer composition; most NPs were successfully endocytosed in the experiments. It is possible that because of the reduced interactions with the membrane, these NPs might disturb the membrane to a lesser extent. According to a previous report, spherical NPs with a size of 200 nm or less were endocytosed by clathrin-coated pits, whereas larger NPs were taken up by caveolae-mediated endocytosis [71]. Anionic NPs can be internalized without membrane interactions, unlike cationic NPs [72]. The NPs described here were successfully delivered to the cells, presumably because of their small size and negative charge. Although these NPs might have a lower cellular uptake than that of cationic NPs, it is likely that they have more potential and applicability as nanocarriers, considering their lower protein corona, longer circulation time, and reduced cytotoxicity/immune response. NPs composed of PLGA and PHA are of great interest because of their physicochemical properties (size, zeta potential, and biodegradation), their capacity for drug encapsulation, and their ability to regulate and steadily release the drug. 

## 5. Conclusions

In conclusion, biodegradable and biocompatible NPs for the delivery of anticancer drugs were fabricated. NPs composed of PLGA and PHA show potential as drug carriers in the following aspects: they are of an appropriate size, have high entrapment efficiency, and offer a sustained-release profile. The findings demonstrate that the use of biocompatible PHA provides advantages over PLGA-based NPs, overcoming some of the drawbacks of the latter, including low encapsulation efficiency and irregular release of PTX. The use of PHA has an important role in controlling drug release and maintaining biocompatibility, as well as biodegradability. The PHA could be applied for pharmaceutical formulations of other hydrophobic chemotherapeutic drugs that require controlled and sustained release. Overall, PHA could offer a wide range of applications for the future design of NP-based drug delivery.

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
