# Peer review of "Polyhydroxyalkanoate Decelerates the Release of Paclitaxel from Poly(lactic-co-glycolic acid) Nanoparticles"

_pharmaceutics, 2022, doi:10.3390/pharmaceutics14081618_

Round 1

Reviewer 1 Report

1. Avoid using we, our in abstract and manuscript

2. Improve the abstract can be done to reflect method, result and conclusion. 

3. In introduction include paclitaxel literature, specially work already done. (Focus on nanoformulations or targeting work done). 

4. Line 106 and 121, specify  the temperature, avoid room temperature 

5. Why only two ratio studied, PLGA:PHA (1:1) (1:9), if all other ratio included study can be more justify the biodegradation

6. Combine discussion along with result for better understanding. 

7. Discussion conclusion must focus and novel out come of study. 

Author Response

Attached is the reponses to Reviewer 1. Please see the attachment (pdf file).

Reviewer 2 Report

Dear Authirs, please edit references and add some from 2022. Also, add conclusion to this work

thank you

Author Response

Attached is the reponses to Reviewer 2. Please see the attachment (pdf file).

Reviewer 3 Report

The manuscript by Si Yeong Lee et al., entitled "Polyhydroxyalkanoate decelerates the release of paclitaxel from 2 poly(lactic-co-glycolic acid) nanoparticles", presents a well-conducted and structured study on four types of nanoparticles loaded with paclitaxel.

The authors prepared biodegradable nanoparticles based on PLGA and PHA, then evaluated their physicochemical properties. Also, they determined the in vitro release rate of the active ingredient and the intracellular uptake of NPs.

Still, some remarks must be addressed:

1. The introduction doesn’t offer any data on the active ingredient used as the "model therapeutic agent". The authors should provide the essential physicochemical data on paclitaxel, useful for a better understanding of the study and also the reasons for its selection in the present research.

2. The authors stated they conducted a storage stability study, but it was performed during one week only and it’s not representative of the real storage conditions.

3. The physicochemical characterization must be carried out in comparison to the raw materials, in order to allow a better view and interpretation of the results.

Author Response

Attached is the reponses to Reviewer 3. Please see the attachment (pdf file).

Reviewer 4 Report

1-    The logicality of the manuscript rationale has to be declared intensively. The novelty of the present work is low. 

2-    In-vitro stability study in saline and rat serum is needed to all formulas and discussed in relation to zeta potential values

3-    What is the localization mechanism predicted for these different formulas in relation to particle sizes more than 200 nm?

4-    The language needs improvement. 

5-    More details of the preparation of different formulas should be presented clearly, step by step. Was a purification processing was conducted?

6-    More details of the characterization process need to be stated. For example, some necessary details of the devices, such as the model and manufacturer, should be presented. The authors could also list some referenced documents here.

7-    TEM images and DLS images are needed.

8-    What are the pH and conductivities of the prepared formulas? How could they guarantee the stability within the subsequent experimental period?

9-    The constituents and concentrations of all formulas compositions might be very important to the physicochemical property and excellent bio-activity of such novel nanoparticles system.

10- Please improve the quality of the figures.

Author Response

Attached is the reponses to Reviewer 4. Please see the attachment (pdf file).

Round 2

Reviewer 3 Report

The authors provided complete answers.